# A Mixed Gas Component Identification and Concentration Estimation Method for Unbalanced Gas Sensor Array Samples

**DOI:** 10.3390/s25196254

**Published:** 2025-10-09

**Authors:** Yuheng Lin, Jinlong Shi, Wanyu Xia, Mingjun Zhou, Yunpeng Gao

**Affiliations:** 1Harbin University of Science and Technology, Harbin 150080, China; 2The Fourth Nineteenth Research Institute of China Electronics Technology Group Corporation, Harbin 150001, China

**Keywords:** gas sensor array, kernel principal component analysis, ADASYN, MLSSVR, sample expansion

## Abstract

Component identification and concentration estimation of a gas mixture component are important for gas detection. However, the accuracy of traditional gas identification will decrease if the sample is not balanced or the number of samples is too few. In this paper, a method based on sample expansion is proposed to solve the aforementioned problem. Firstly, the ADASYN-ELM method is proposed to identify the composition of a gas mixture component. The KPCA is used to extract the feature of the sensor signal and the ADASYN method is used to expand the samples. The PSO and GA algorithms were used to optimize the parameters of the ELM classification model to complete the qualitative analysis. Secondly, the S-SMOTE-MLSSVR method was put forward to quantitatively estimate. The S-SMOTE method was used to expand the samples, and the regression model MLSSVR was optimized by PSO and GA algorithms to complete the quantitative analysis. The results show that the accuracy rate after sample expansion is generally higher and the MAPE and RMSE are generally lower than before sample expansion, indicating that the sample expansion method has a positive effect on classification and concentration estimation of mixed gases with extremely unbalanced samples and too few samples.

## 1. Introduction

Various gases usually exist in the form of mixtures in the industrial and living environment; it is particularly important to identify the composition and concentration analysis of the gas mixture component, so the detection of the gas mixture component has become a hot research field. There are three methods for gas detection; the first method is sensory evaluation, which is an evaluation method based on the reaction of human sensory organs to gas. However, sensory evaluation is susceptible to the limitations of the human subjective and olfactory system, and some gases can also cause damage to the human body. The second method is chemical analysis, such as spectroscopy, gas chromatography and mass spectrometry; this method uses advanced chemical analysis equipment to measure the composition and concentration of gases. However, the difficulty of sampling, the complexity of operation, the high cost of equipment and the low real-time performance of these methods limit the application of these methods to some extent. The last method is to use gas sensors to detect gases. This method has a low cost and is easy to operate. However, in complex environments or when there are many types of gases, it is difficult to detect the gases using a single sensor. In addition, other influencing factors such as the sensitivity of sensing materials to target gases, the reproducibility of sensor arrays and limitations in the quantity of discriminated gas all impose limitations on the application of the sensor [1,2,3].

Due to the limitations of the above methods, Gardner published a review article on electronic nose, formally proposing the concept of “electronic nose” in 1994 [4]. Electronic nose, also known as artificial olfactory system, is a new bionic detection technology which can be used to simulate the working mechanism of biological olfactory. It uses sensor array technology, which has the advantages of fast response, high sensitivity, low cost and easy processing. Electronic nose technology not only solves the subjective problems of sensory analysis, but also overcomes the complicated and expensive problems of chemical analysis methods. Electronic nose technology is often used in the analysis and detection of various gas fields, such as pollution control [5,6], medical technology [7,8], oil exploration [9], food safety [10,11], agricultural science [12,13] and environmental science [14,15], and the application of electronic noses is very extensive [16,17,18]. Therefore, electronic nose technology has become an important research direction in the field of gas detection. The electronic nose system is mainly composed of gas sensor array, signal preprocessing module and pattern recognition, as shown in Figure 1.

The sensor array is composed of various types of sensors and converts chemical signals into electrical signals by converting A/D. The signal preprocessing is mainly to remove noise, extract feature and process signal of sensor response signal. The pattern recognition is the classification recognition and concentration estimation of measured gases using machine learning algorithms. At present, the key research directions of electronic nose are mainly the performance improvement of gas sensors and the research of various machine learning methods in gas sensors [19]. Due to the nonlinear response of MOS gas sensor, it is difficult to improve mixed gas classification and concentration prediction by simply relying on the selection of gas-sensitive materials. Therefore, appropriate machine learning algorithms are needed to solve the current problems [20]. Machine learning algorithm plays an extremely important role, and its accuracy, time efficiency and anti-interference ability all affect the decision result. Among the research results of many studies, the literature [21] argues that more intelligent pattern recognition technology is needed to realize the potential of electronic nose technology. Literature [22] proposes that reasonable improvement of algorithms is an important support for the development of machine olfaction.

Before pattern recognition, data sets need to be preprocessed and feature extraction, which is conducive to providing reasonable data sets for subsequent recognition models. The main steps of data preprocessing are data cleaning, data specification and data transformation. Data cleansing is the “cleaning” of data by filling in missing values, smoothing noise data, smoothing or removing outliers, and resolving data inconsistencies. The average method can solve the problem of missing data, and the averaging method is to fill the average value of all data into the data position to be compensated [23]. Based on the setting principle, the nearest location data is selected as the compensation data [24]. The compensation method of regression model is to build a regression model and fill the compensation position with the predicted value of the regression model as the compensation value [25]. Although the data specification technique is represented by a much smaller data set, it still maintains close data integrity and can be mined on the data set after the specification, and it is more efficient and produces nearly identical analysis results. Common strategies are dimension specification and dimension transformation. Common loss dimension transformation methods include principal component analysis (PCA), linear discriminant analysis (LDA), singular value decomposition (SVD), etc. [26,27]. Data transformation includes normalization, discretization and sparse processing of data to achieve the purpose of mining. In the literature [28], data standardization and baseline processing methods were used as data processing methods to realize the identification of nitrogen dioxide and sulfur dioxide in air pollution. The feature extraction methods include PCA and LDA. When the data is small, the feature extraction effect of PCA is better than that of LDA, but when the data is large, the feature extraction effect of LDA is better than that of PCA [29].

In the classification study of mixed gases, researchers mostly adopt machine learning methods [30,31,32,33,34], and the selection of classification algorithms needs to be based on the characteristics of samples to find a more suitable scheme [35,36,37,38,39,40,41]. Sunny used four thin film sensors to form a sensor array to identify and estimate the concentration of a gas mixture. PCA was used to extract the features of response signals, and ANN and SVM were used for category recognition, achieving good recognition results [42]. Zhao studied the recognition of gas mixture components of organic volatiles. An array composed of four sensors was used to identify formaldehyde when the background gas was acetone, ethanol and toluene. PCA was used for dimensionality reduction extraction. MLP, SVM and ELM are, respectively, used in the classifier to identify and classify, among which SVM achieves the best effect and ELM requires the shortest training time to obtain results [43]. Jang used the SVM and paired graph scheme combined with the sensor array composed of semiconductor sensors to classify CH4 and CO, and obtained a high recognition accuracy [44]. Jung used sensor array to collect gas and then used SVM and fuzzy ARTMAP network for experimental comparison. The recognition time of SVM was shorter than the fuzzy ARTMAP network [45]. Zhao adopted a weighted discriminant extreme learning machine (WDELM) as a classification method. WDELM assigns different weights to each specific sample by using a flexible weighting strategy, which enables it to perform classification tasks under unbalanced class distribution [46].

There are multiple regression [47], neural network [48,49], SVR [50] and other methods [51,52,53,54,55,56] for gas concentration analysis. Zhang used the WCCNN-BiLSTM model to automatically extract time–frequency domain features of dynamic response signals from the original signals to identify unknown gases. The time domain characteristics of the steady-state response signal are automatically extracted by the many-to-many GRU model to accurately estimate the gas concentration [57]. Piotr proposed an improved cluster-based ensemble model to predict ozone. Each improved spiking neural network was trained on a set of separate time series; the ensemble model could provide better prediction results [58]. Liang used AdaBoost classification algorithm to classify the local features of infrared spectrum, and carried out PLS local modeling according to different features to predict the concentration of a gas mixture component. This method solves the problems of difficult identification and inaccurate quantitative analysis of alkane mixture gas components in traditional methods [59]. Adak used the multiple linear regression (MVLR) algorithm to predict the concentration of a mixture of two gases in acetone. The relative errors of acetone and methanol are lower than 6% and 17%, respectively [60]. 

Based on the above literature, most of the methods are suitable for relatively balanced data sets of categories. When the number of samples is extremely unbalanced, the traditional method with the overall classification accuracy as the learning objective will pay too much attention to most categories. As a result, the classification or regression performance of a small number of class samples is degraded, which leads to the failure of traditional machine learning to work well on extremely unbalanced data sets. Secondly, PCA method is often used to solve linear problems in the literature, while most of the problems in the real environment are nonlinear problems. In addition, ML algorithms often have multi-parameter and difficult parameter to determine the problem. In the literature, the parameters of deep learning algorithms such as neural network or ELM are often obtained by trial and error method or experience, and the selection of parameters plays a crucial role in the performance evaluation of algorithms. When the learning algorithm model without optimal parameters is used to detect the mixed gas, it cannot reasonably compare other algorithms with evaluation criteria. To solve the above problems, this paper presents a gas mixture component detection method which is suitable for electronic nose under unbalanced conditions. In view of the problem of extremely unbalanced sample numbers and too few samples, SMOTE, ADASYN, B-SMOTE, S-SMOTE and CSL-SMOTE were put forward for artificial synthesis of new samples by sample expansion methods, so as to alleviate the problem. For nonlinear problems, Kernel Principal Component Analysis (KPCA) method is used for feature extraction, and kernel technique is used to extend PCA to nonlinear problems. To solve the problem of multi-parameter and difficult parameter determination, PSO and GA optimization methods are used to optimize the parameters of classification and regression models, which are convenient for classification and regression methods to identify and classify the mixed gas and estimate the concentration.

The rest of the paper is structured as follows. In Part II, the methods of feature extraction, sample expansion, classification recognition and concentration detection are briefly introduced. In Part III, a new method for detecting mixed gas based on the electronic nose is introduced in detail. The verification experiment is carried out in part IV, and the experimental results are analyzed and discussed. Part V is the summary and outlook.

## 2. Methods

### 2.1. Kernel Principal Component Analysis

The KPCA transforms the linearly indivisible sample input space into the divisible high-dimensional feature space through kernel function $\Phi \left(\cdot \right)$ and performs PCA in this high-dimensional space. Compared with the linear problem solved by PCA, the KPCA with kernel technique extends the linear problem to the nonlinear problem [12].

Set $X=[x_{1},x_{2},\cdots ,x_{N}]\in R^{M\times N}$ as the observation sample after pretreatment, and contains N samples in X, $x_{i}\in R^{M}$ represents the ith observed sample of the M dimension. The covariance matrix mapping sample X to a high dimensional feature space is expressed as(1)$$C=\frac{1}{N}\Phi \left(X\right){\left[\Phi \left(X\right)\right]}^{T}=\frac{1}{N}\sum_{i=1}^{N}\Phi \left(x_{i}\right)\Phi {\left(x_{i}\right)}^{T}$$
where $\sum_{i=1}^{N}\Phi \left(x_{i}\right)=0$, $\Phi \left(\cdot \right)$ is a nonlinear mapping.

Eigenvalue decomposition of covariance matrix C:(2)$$\lambda v=Cv=\frac{1}{N}\Phi \left(X\right)\Phi {\left(X\right)}^{T}v=\frac{1}{N}\sum_{i=1}^{N}\Phi \left(x_{i}\right)\Phi {\left(x_{i}\right)}^{T}v$$
where λ and v, respectively, represent the eigenvalues and eigenvectors of covariance matrix C, and v is the eigenvector in the eigenspace, that is, the direction of the principal component. There is a coefficient vector $\alpha ={\left(\alpha_{1},\alpha_{1},\cdots ,\alpha_{N}\right)}^{T}$ for a linear representation of the eigenvector v:(3)$$v=\sum_{i=1}^{N}\alpha_{i}\Phi \left(x_{i}\right)=\Phi \left(X\right)\alpha$$

Substitute (3) into (2) and multiply both ends by $\Phi {\left(X\right)}^{T}$ to obtain the following equation:(4)$$\lambda \left[\Phi {\left(X\right)}^{T}\right]\Phi \left(X\right)\alpha =\frac{1}{N}{\left[\Phi \left(X\right)\right]}^{T}\Phi \left(X\right){\left[\Phi \left(X\right)\right]}^{T}\Phi \left(X\right)\alpha$$

Define $K=[\Phi {\left(X\right)}^{T}]\Phi (X)$, then K is the symmetric positive semidefinite matrix of N×N:(5)$$K_{ij}=K\left(x_{i},x_{j}\right)={\left[\Phi \left(x_{i}\right)\right]}^{T}\Phi \left(x_{j}\right)$$
where $K_{ij}$ represents the elements in row i and column j of matrix K, and the eigenvalue solution problem combined with Equations (3)–(5) is converted to(6)Nλα=Kα
where Nλ is the eigenvector of K, and principal component analysis (PCA) is performed in the eigenspace to solve the eigenproblem of Formula (6), and the eigenvalue $\lambda_{1}\geq \lambda_{2}\geq \cdots \geq \lambda_{N}$ corresponding to the eigenvector $\alpha_{1},\alpha_{2},\cdots ,\alpha_{N}$ is obtained;(7)$$r_{CCR}=\sum_{i=1}^{P}\lambda_{i}/\sum_{j=1}^{N}\lambda_{j}\times 100\%$$
where p is the number of primary components.

The kth feature of the newly observed sample x is mapped by $\Phi \left(x\right)$ to $v_{k}$, where $v_{k}$ is the feature vector of the kth feature in the feature space, i.e., the direction of the principal component.(8)$$t_{k}=\langle v_{k},\Phi \left(x\right)\rangle =\sum_{i=1}^{N}{\alpha_{i}}^{k}\langle \Phi \left(x_{i}\right),\Phi \left(x\right)\rangle \;\;\;\;k=1,2,\cdots ,p$$
where $t_{k}$ is the projection of $\Phi \left(x\right)$ onto $v_{k}$. Where $\sum_{k=1}^{N}\Phi \left(x_{k}\right)=0$ is not satisfied, K is $\tilde{K}$:(9)$$\tilde{K}=K-I_{N}K-KI_{N}+I_{N}KI_{N}$$
where $\tilde{K}$ is the kernel matrix after centralization, and $I_{N}$ is the matrix of N×N, where each element is 1/N.

### 2.2. The Safe-Level-SMOTE Method

The SMOTE method can alleviate the over-fitting problem caused by random oversampling, but it only considers a few types of cases and does not consider the overlap between the synthesized samples and most types of samples; therefore, most researchers tend to adopt the improved Safe-Level-SMOTE method [61,62,63,64]. The Safe-Level-SMOTE method will select a few classes with a high degree of safety and assign a certain degree of safety to each class separately before combining new classes, which will be closer to the high degree of safety. This method solves the quality problem of the SMOTE class and the problem of fuzzy class boundaries. The schematic diagram of a few class samples synthesized by the Safe-Level-SMOTE algorithm is shown in Figure 2.

The process of the Safe-Level-SMOTE method is as follows:(1)Find the k nearest neighbors of p, denoting the number of k neighbors in D as slp, and denoting a certain neighbor as n. (2)Find the k nearest neighbors of n, and the number of k neighbors in D is denoted as sln.(3)Set the ratio sl_ratio=slp/sln.

where D is a sample set of a few classes, and p is a sample in D. 

Case 1: sl_ratio=∞ and slp=0, that is, the k neighbors of the minority class sample p are all majority class samples, and no composite data is generated in this case.

Case 2: sl_ratio=∞ and slp≠0, that is, when sln is very small relative to slp, the ratio will be 0. The n sample point is located in most class samples, and then p point is copied.

Case 3: sl_ratio=1, that is, slp=sln, at this time to synthesize a new sample between n and p, synthesis method same as smote.

Case 4: sl_ratio>1, that is, slp>sln, at this time, the number of subclass samples around p point sample is greater than the number of subclass samples around n point sample, consider p point as the safe level, and use the smote in β=0∼1/sl_ratio between p and n point to synthesize a new sample, and the synthesized sample position is biased to p point.

Case 5: sl_ratio<1, that is, slp<sln then the number of subclass samples around p point sample is less than the number of subclass samples around n point sample, consider n point as the safe level, between p and n point with β=1−sl_ratio∼1 to synthesize a new sample, the synthesized sample position is biased to n point.

### 2.3. Adaptive Synthetic Sampling Approach (ADASYN)

Many classification problems will face the problem of sample imbalance, most of the algorithms in this case, and classification effect is not ideal. Researchers usually adopt the SMOTE method to address the issue of sample imbalance. Although the SMOTE algorithm is better than random sampling, it still has some problems. Generating the same number of new samples for each minority class sample may increase the overlap between classes and create valueless samples. Therefore, the improved ADASYN method of SMOTE [65] is adopted. The basic idea of this algorithm is to adaptively generate minority class samples based on their distribution. This method can not only reduce the learning bias caused by class imbalance but also adaptively shift the decision boundary to the difficult-to-learn samples. Then, new samples are artificially synthesized based on the minority class samples and added to the data set.

The process of the ADASYN method is as follows:

Input: Training data set $D_{tr}$ with m samples: $\left\{x_{i},y_{i}\right\}\left(i=1,2,\cdots m\right)$, where $x_{i}$ is a sample of n dimensional feature space X, $x_{i}$ corresponds to class label $y_{i}\in Y=\left\{-1,1\right\}$. $m_{s}$ and $m_{l}$ are defined as minority sample size and majority sample size, respectively, so $m_{s}\leq m_{l}$ and $m_{s}+m_{l}=m$.

The algorithm process:(1)Calculate the unbalance degree:(10)$$d=m_{s}/m_{l},d\in \left(0,1\right]$$(2)If $d<d_{th}$($d_{th}$ is the default threshold of the maximum allowable unbalance rate):(a)Calculate the amount of composite samples that need to be generated for a few classes of samples:(11)$$G=\left(m_{l}-m_{s}\right)\times \beta$$
where $\beta \in \left[0,1\right]$ is a parameter that specifies the level of balance required after the resultant data is generated, and β=1 indicates that the new data set is completely balanced after the resultant.(b)For each $x_{i}$ belonging to the minority class, find K neighbors based on Euclidean distances in n dimensional space, and compute the ratio $r_{i}$, which is defined as(12)$$r_{i}=\Delta_{i}/K,i=1,2,\cdots m_{s}$$
where $\Delta_{i}$ is a sample of most classes input in K neighbors, then $r_{i}\in \left[0,1\right]$.(c)Normalizes $r_{i}$ according to ${\hat{r}}_{i}=r_{i}/\sum_{i=1}^{m_{s}}r_{i}$, so $r_{i}$ is a density distribution ($\sum_{i}{\hat{r}}_{i}=1$).(d)Calculate the amount of sample $x_{i}$ that needs to be synthesized in each minority sample:(13)$$g_{i}={\hat{r}}_{i}\times G$$
where G is the total sample size of artificial minority samples synthesized according to Formula (11).(e)For each minority sample $x_{i}$, the sample $g_{i}$ is synthesized by following the following steps:Do the Loop from 1 to $g_{i}$;(f)(i) Randomly select a minority sample $x_{zi}$ from K neighbors of $x_{i}$; (ii) Synthetic sample $s_{i}=x_{i}+\left(x_{zi}-x_{i}\right)\times \lambda$; where $\left(x_{zi}-x_{i}\right)$ is a difference vector in n dimensional space and $\lambda \in \left(0,1\right]$ is a random number.End Loop

As can be seen from the above steps, the key idea of the ADASYN method is to use density distribution as a criterion to adaptively synthesize the number of artificial samples for each minority class sample. From a physical perspective, the distribution of weights is measured based on the learning difficulty of different minority class samples. The data set obtained by the ADASYN method not only solves the problem of imbalanced data distribution (according to the expected balance level defined by the β coefficient), but also forces the learning method to focus on those difficult-to-learn samples.

### 2.4. The Multi-Output Least Squares Support Vector Regression Machine (MLLSVR)

Support vector regression machine (SVR) is a traditional machine learning method for solving convex quadratic programming problems. The basic idea of the method is to map the input vector to a high-dimensional feature space through a pre-determined nonlinear mapping, and then perform linear regression in this space. Thus, the effect of nonlinear regression in the original space is obtained [19]. Least squares support vector regression machine is an improved version that replaces inequality constraints in SVR with equality constraints. MLSSVR is a generalization of LSSVR in the case of multiple outputs.

Suppose data set ${\left\{\left(x_{i},y_{i}\right)\right\}}_{i=1}^{l}$, where $x_{i}\in \Omega^{n}$ is the input vector and $y_{i}\in \Omega$ is output value. Nonlinear mapping $\varphi :\Omega^{n}\to \Omega^{n_{h}}$ is introduced to map input to the $n_{h}$ dimensional feature space, and the regression function is constructed.(14)$$f\left(x\right)=\varphi {\left(x\right)}^{Τ}w+b$$
where $w\in \Omega^{n_{h}\times l}$ is the weight vector and b∈R is the offset coefficient.

In order to find the best regression function, the minimum norm $\left\|w\right\|=w^{T}w$ is needed. The problem can be boiled down to the following constraint optimization problem:(15)$$\begin{array}{c} \underset{w\in R^{n_{h}\times l},b\in R}{\min}S\left(w\right)=\frac{1}{2}w_{0}^{Τ}w_{0}\\ s.t.y=Z^{T}W+b1_{l} \end{array}$$
where $\;y\in \Omega^{l\times l}$ is a block vector composed of $y_{i}$, $Z=\left(\varphi \left(x_{1}\right),\varphi \left(x_{2}\right),\cdots ,\varphi \left(x_{l}\right)\right)\in \Omega^{n_{h}\times l}$, $1_{l}={\left(1,1,\cdots ,1\right)}^{T}\in \Omega^{l\times 1}$. By introducing the relaxation variable $\xi_{i}\in \Omega \left(i\in N_{l}\right)$, the minimization problem of Equation (15) can be transformed into(16)$$\begin{array}{l} \underset{w\in \Omega^{n_{h}\times l},b\in \Omega }{\min}J\left(w,\xi \right)=\frac{1}{2}\left(w^{T}w\right)+\frac{\gamma }{2}\xi^{Τ}\xi \\ s.t.y=Z^{T}W+be_{l}+\xi \end{array}$$
where $\xi ={\left(\xi_{1},\xi_{2},\cdots ,\xi_{l}\right)}^{T}\in \Omega^{l}$ is a vector composed of relaxation variables and $\gamma \in \Omega_{+}$ is a regularization parameter.

In the multiple-output case, for a given training set ${\left\{\left(x_{i},y_{i}\right)\right\}}_{i=1}^{l}$, $x_{i}\in \Omega^{n}$ is the input vector, $y_{i}\in \Omega^{m}$ is the output vector, $X\in \Omega^{l\times n}$ and $Y\in \Omega^{l\times m}$ are composed of block matrices of $x_{i}$ and $y_{i}$, respectively. The purpose of MLSSVR is to map from n dimensional input $x_{i}\in \Omega^{n}$ to m dimensional output $\;y_{i}\in \Omega^{m}$. As in the case of single output, the regression function is(17)$$f\left(x\right)=\varphi {\left(x\right)}^{T}W+b^{T}$$
where $W=\left(w_{1},w_{2},\cdots ,w_{m}\right)\in \Omega^{n_{h}\times m}$ is a matrix composed of weight vectors and $b=\left(b_{1},b_{2},\cdots ,b_{m}\right)\in \Omega^{m}$ is a vector composed of offset coefficients. Minimize the following constrained objective function by finding W and b:(18)$$\begin{array}{l} \underset{W\in \Omega^{n\times_{hm}},b\in \Omega^{m}}{\min}J\left(W,\Xi \right)=\frac{1}{2}trace\left(W^{Τ}W\right)+\frac{\gamma }{2}trace\left(\Xi^{Τ}\Xi \right),\\ s.t.Y=Z^{T}W+\mathrm{repmat}\left(b^{Τ},l,1\right)+\Xi \end{array}$$
where $\Xi =\left(\xi_{1},\xi_{2},\cdots ,\xi_{m}\right)\in \Omega_{+}^{l\times m}$ is a matrix of relaxation vectors. By solving this problem, W and b are obtained, and the nonlinear mapping is obtained. According to hierarchical Bayes, the weight vector $w_{i}\in \Omega^{n_{h}\times 1}\left(i\in {Ε}_{m}\right)$ can be decomposed into the following two parts:(19)$$w_{i}=w_{0}+v_{i}$$
where $w_{0}\in \Omega^{n_{h}\times 1}$ is the mean vector, $v_{i}\left(i\in \Omega_{m}\right)$ is a difference vector, $w_{0}$ and $v_{i}\left(i\in {Ε}_{m}\right)$ reflect the connectivity difference between outputs. That is, $w_{0}$ contains the general characteristics of output, and $v_{i}\left(i\in {Ε}_{m}\right)$ contains the special information of ith component of the output. Equation (18) is equivalent to the following problem.(20)$$\begin{array}{l} \underset{W_{0}\in \Omega^{n_{h}},V\in \Omega^{n_{h}\times m},b\in \Omega^{m}}{\min}S\left(w_{0},V,\Xi \right)=\frac{1}{2}w_{0}^{Τ}w_{0}+\frac{\lambda }{2m}trace\left(V^{Τ}V\right)+\gamma \frac{1}{2}trace\left(\Xi^{Τ}\Xi \right),\\ s.t.Y=Z^{T}W+\mathrm{repmat}\left(b^{Τ},l,1\right)+\Xi \end{array}$$
where $V=\left(v_{1},v_{2},\cdots ,v_{m}\right)\in \Omega^{n_{h}\times m}$, $W=\left(w_{0}+v_{1},w_{0}+v_{2},\cdots ,w_{m}+v_{m}\right)\in \Omega^{n_{h}\times m}$, $Z=\left(\varphi \left(x_{1}\right),\varphi \left(x_{2}\right),\cdots ,\varphi \left(x_{l}\right)\right)\in \Omega^{\Omega^{n_{h}\times l}}$, $\lambda ,\gamma \in \Omega_{+}$ are two regularization parameters. 

The Lagrange function corresponding to Equation (20) is(21)$$L\left(w_{0},V,b,\Xi ,A\right)=J\left(w_{0},V,\Xi \right)-trace(\;A^{Τ}(Z^{T}W+\mathrm{repmat}\left(b^{T},l,1\right)+\Xi -Y))$$
where $A=\left(\alpha_{1},\alpha_{2},\cdots \alpha_{m}\right)\in \Omega^{l\times m}$ is a matrix consisting of a Lagrange multiplier vector.

According to the optimization theory of Karush–Kuhn–Tucker (KKT) conditions, the linear following equations are obtained:(22)$$\left\{\begin{array}{l} \frac{\partial L}{\partial w_{0}}=0\Rightarrow w_{0}=\sum_{i=1}^{m}Z\alpha_{i}\\ \frac{\partial L}{\partial V}=0\;\Rightarrow V=\frac{m}{\lambda }ZA\\ \frac{\partial L}{\partial b}=0\Rightarrow A^{Τ}1_{l}=0_{l}\\ \frac{\partial L}{\partial \Xi }=0\;\Rightarrow A=\gamma \Xi \\ \frac{\partial L}{\partial A}=0\;\Rightarrow Z^{T}W+\mathrm{repmat}\left(b^{T},l,1\right)+\Xi -Y=0_{l\times m} \end{array}\right.$$

By canceling W and Ξ in Equation (22), the linear matrix equation can be obtained as follows:(23)$$\left[\begin{array}{ll} 0_{ml\times m} & P^{Τ}\;\\ P & H \end{array}\right]\left[\begin{array}{l} b\\ \alpha \end{array}\right]=\left[\begin{array}{l} 0_{m}\\ y \end{array}\right]$$
where $P=blockdiag\left(\overset{m}{\overset{︷}{1_{l},1_{l},\cdots ,1_{l}}}\right)\in \Omega^{ml\times m}$, $H=\Omega +\left(1/\gamma \right)I_{ml}+\left(m/\lambda \right)Q\in \Omega^{ml\times ml}$, $K=Z^{Τ}Z\in \Omega^{l\times l}$, $\Omega =repmat\left(K,m,m\right)\in \Omega^{ml\times ml}$, $Q=blockdiag\left(\overset{m}{\overset{︷}{K,K,\cdots ,K}}\right)\in \Omega^{ml\times ml}$, $\alpha ={\left(\alpha_{1}^{Τ},\alpha_{2}^{Τ},\cdots ,\alpha_{m}^{Τ}\right)}^{Τ}\in \Omega^{ml}$ and $y={\left(y_{1}^{Τ},y_{2}^{Τ},\cdots ,y_{m}^{Τ}\right)}^{Τ}\in \Omega^{ml}$. Since H is not positive definite, Equation (23) can be changed to the following form:(24)$$\left[\begin{array}{ll} \;S & 0_{ml\times ml}\\ 0_{m\times m} & H \end{array}\right]\left[\begin{array}{c} b\\ H^{-1}Pb+\alpha \end{array}\right]=\left[\begin{array}{c} P^{Τ}H^{-1}y\\ y \end{array}\right]$$
where $S=P^{Τ}H^{-1}P\in \Omega^{m\times m}$ is a positive definite matrix. It is not difficult to see that Formula (24) is positive definite. The solution $b^{*}$ and $\alpha^{*}$ of Equation (23) are obtained in three steps:

(1) Solve: η, μ from Hη=P and Hμ=y; (2) calculate: $S=P^{Τ}\eta$; (3) solve: $b^{*}=S^{-1}\eta^{Τ}y$, $\alpha^{*}=\mu -\eta b$.

The corresponding regression function can be obtained as follows:(25)$$\begin{array}{l} f\left(x\right)=\varphi {\left(x\right)}^{Τ}W^{*}+b^{*Τ}\\ =\varphi {\left(x\right)}^{Τ}repmat\left(w_{0}^{*},1,m\right)+\varphi {\left(x\right)}^{Τ}V^{*}+b^{*Τ}\\ =\varphi {\left(x\right)}^{Τ}repmat\left(\sum_{i=1}^{m}Z\alpha_{i}^{*},1,m\right)+\frac{m}{\lambda }\varphi {\left(x\right)}^{Τ}Z^{*}+b^{*Τ} \end{array}$$

This article uses the most common RBF kernel functions, as follows.(26)$$k\left(x,x_{j}\right)=\exp\left(-g{\left\|x-x_{j}\right\|}^{2}\right)$$
where $g=1/2\sigma^{2}$, $\sigma \in \Omega^{+}$ is the kernel width.

The MIMO differs from MISO algorithm in input–output mapping system and parameter types. When using this method, an optimization algorithm is needed to optimize the parameters in its model.

## 3. The Improvement Method

This paper proposes a method of mixture gas identification and concentration detection based on sample expansion. The flow chart of gas identification and concentration detection using the proposed method is shown in Figure 3.

The qualitative analysis of gas mixture is divided into five steps: data preprocessing, feature extraction, stratified cross-validation, sample expansion, parameter optimization and qualitative identification.

Step 1: The raw signal is preprocessed to eliminate the difference caused by the baseline to the raw data.

Step 2: The KPCA is used to extract the features of the preprocessed signal. When the cumulative contribution rate of n feature values reaches the set threshold, the first n features are selected to represent the original features.

Step 3: After KPCA feature extraction, use hierarchical five-fold cross-validation to divide the data into five mutually exclusive subsets on average. In each experiment, one subset is selected as the test set, the other four subsets are combined as the training set, and the average of the five results is used as the estimation of the algorithm accuracy.

Step 4: In the training set, the ADASYN method is used to artificially synthesize a few class samples in the class imbalance, and the generated new samples are put into the training set to form a new training set.

Step 5: After sample expansion on the class unbalanced data set, The ELM method is adopted as the classification method, the PSO and GA are used to optimize the parameters of the classification method, and the classification model is obtained. The test set is input into the classification model to identify the gas mixture.

The quantitative analysis of a gas mixture component is divided into four steps: data preprocessing, sample expansion, parameter optimization and quantitative estimation.

Step 1: The original signal is preprocessed to eliminate the influence of the baseline.

Step 2: Arrange the concentration in ascending order in the pre-treated sample set, and cross-select the samples as the training set and the test set. In the training set, The S-SMOTE method was used to synthesize artificial samples. The generated samples are put into the training set to form a new training set.

Step 3: After sample expansion, The MLSSVR method is used as the regression, and the PSO and GA methods are used to optimize the parameters of the regression method, and the regression model with the optimal parameters is obtained.

Step 4: Input the test set into the regression model to obtain the estimation of the mixed gas concentration, and use the mean absolute percentage error (MAPE) and root mean square error (RMSE) as the evaluation criteria.

## 4. The Experiment and Results

### 4.1. The Experimental Platform and Data Set

In order to verify the validity of a sample augmentation-based mixture gas identification and concentration detection method proposed in this paper, validation experiments were performed on UCI publicly available data sets. The data set was collected at the Gas Delivery Platform facility in the Chemical Signals Laboratory at the Bio-circuits Institute at the University of California San Diego. The system consists of data acquisition system platform, power control module and chemical delivery system. The sensor array includes 16 chemical sensors of four different types (Figaro Company, Meadows, IL, USA): TGS-2600, TGS-2602. TGS-2610. TGS-2620 (each type with four units). These sensors are integrated with custom signal conditioning and control electronic equipment. During the entire experiment, the working temperature of the sensors is controlled, and the working voltage of the sensors remains at 5 V. The sensor array continuously obtains electrical conductivity at a sampling frequency of 100 Hz. The sensor array is placed in a 60 mL measurement chamber, and the gas sample is injected at a constant flow rate of 300 mL/min. The flow control system is based on three different branch polyethylene. To separately control the gas flow of each branch while maintaining the total flow constant, each branch is connected to a different pressurized gas cylinder through a mass flow controller (MFC) system. The first fluid branch is used to control the flow of dry air provided by Airgas Company in the pressurized gas cylinder. The other two branches can be freely connected to any pressurized gas cylinder. These three branches converge to obtain the required gas mixture. Finally, the generated mixture is continuously circulated through the measurement chamber and collected by the exhaust system. To obtain accurate and repeatable data generation, this system is fully operated by a computerized environment, as shown in Figure 4.

The data set includes two binary gas mixtures: ethylene and methane, ethylene and CO. In this paper, part of the data set was selected as the data set for study, and 161 samples were selected in total. The composition of the experimental sample set is shown in Table 1. There are five types of gases, CO, ethylene, ethylene and methane mixture, ethylene and CO mixture, methane, labeled with category labels ranging from 1 to 5 and the sample size of each gas type.

### 4.2. The Preprocessing

The relative difference method is adopted, as shown in Formula (27). Baseline correction of signals for differences caused by raw data, compensate for drift in chemical sensor arrays and eliminate noise in sensor responses. The relative difference method eliminates the difference between baseline and raw data; the reliability of the data is ensured.(27)$$T_{i}=\frac{S_{i}-S_{o}}{S_{o}}$$
where $S_{o}$ is the baseline value, $S_{i}$ is the recorded value of the sensor, and $T_{i}$ is the effective value of the sensor.

### 4.3. The Extended Sample

The data set of qualitative analysis is divided into training and test set according to stratified five-fold cross-validation. In the training set, the gas type of gas mixture is labeled, the sensor array response is taken as input, the sample expansion method is used to synthesize the sample artificially, and the new sample is put into the training set to form a new training set. The expansion amounts of different sample expansion methods are shown in Table 2.

The data set of the concentration of the mixed gas is arranged in ascending order, samples are cross-selected as the training set and the test set, and the type of the mixed gas is labeled, the response and concentration of the sensor array are taken as the input, the sample expansion method is used to artificially synthesize samples, and new samples are put into the training set to form a new training set. The expansion amounts of each gas based on different sample expansion methods are shown in Table 3.

### 4.4. The Qualitative Identification

#### 4.4.1. The Hierarchical K-Fold Cross-Validation

Hierarchical five-fold cross-validation is to divide the data into five subsets on average, and the proportion of sample size of each subset is the same as the original data set. Each experiment selects a subset as the test, combines the other four subsets as the training set, and the average of the five results is used as the estimation of the algorithm accuracy. After KPCA feature extraction, the data set is divided into five mutually exclusive subsets (D1~D5) of the same size using hierarchical five-fold cross-validation. In each test, any subset from D1 to D5 is selected as the test set, and the remaining 4 subsets are combined as the training set. If D1 is used as the test set, D2 to D4 are combined as the training set, and the results of the five times are averaged; the five-fold hierarchical cross-validation data allocation is shown in Figure 5.

#### 4.4.2. The Feature Extraction

After preprocessing, the maximum value of sensor effective is selected for each sample to obtain data set. KPCA was used to extract features. In the selection of kernel functions, since the polynomial kernel function and the neural network kernel function have very strict requirements for parameter selection, they are prone to causing ill-conditioned kernel matrices, generating negative eigenvalues and eigenvectors, thereby leading to the failure of the KPCA transformation model construction. In comparison, the radial basis kernel function transformation matrix has excellent positive definiteness and is suitable for a wide range of parameters. Therefore, in this study, the radial basis kernel function was selected as the transformation kernel function. The parameter $\sigma^{2}$ of the radial basis function was default selection. The cumulative contribution threshold was set. When the number of eigenvalues is 5, the cumulative contribution rate reaches more than 95%, indicating the first five principal components can roughly represent all data, as shown in Figure 6. Therefore, the original feature can be characterized by the first five eigenvalues.

In the training set, SMOTE, Borderline-SMOTE (B_SMOTE), Safe-Level-SMOTE (S_SMOTE), Cost-Sensitive Learning-Smote (CSL_SMOTE) and ADASYN were used to expand the samples. The distribution of various data sets with the first three principal components is shown in Figure 7a. The synthetic sample with S-SMOTE method is shown in Figure 7b. The blue part is the synthetic artificial sample of various types. As the figure shows, the newly generated aggregate of various types did not overlap with other classes in their original categories, indicating that the synthetic sample method effectively alleviated the class imbalance. Then the PSO and GA methods were used to optimize the parameters of MRVM, SVM, ELM and SOFTMAX, and the optimal parameter classification model was obtained to identify and classify the mixed gas. The results of MRVM, SVM, ELM and SOFTMAX methods are shown in Table 4, Table 5, Table 6 and Table 7 and Figure 8, Figure 9, Figure 10 and Figure 11. It can be seen from the figures and tables that the accuracy rate after using the expanded sample is higher than that before the expanded sample, indicating that the expanded sample methods are beneficial to the classification of unbalanced gas mixture components.

#### 4.4.3. The Classification of Mixed Gases Based on SOFTMAX Method

The results of SOFTMAX classification method are shown in Table 4 and Figure 8; the K value represents the nearest neighbor.

**Table 4 sensors-25-06254-t004:** The classification of mixed gases based on SOFTMAX method.

SOFTMAX	SMOTE	ADASYN	B_SMOTE	S_SMOTE	CSL_SMOTE
K	5	5	2	5	5
Unexpanded accuracy (%)	93.6	93.55	96.8	90.3	96.8
Expanded accuracy (%)	96.8	100	100	96.8	100

**Figure 8 sensors-25-06254-f008:**
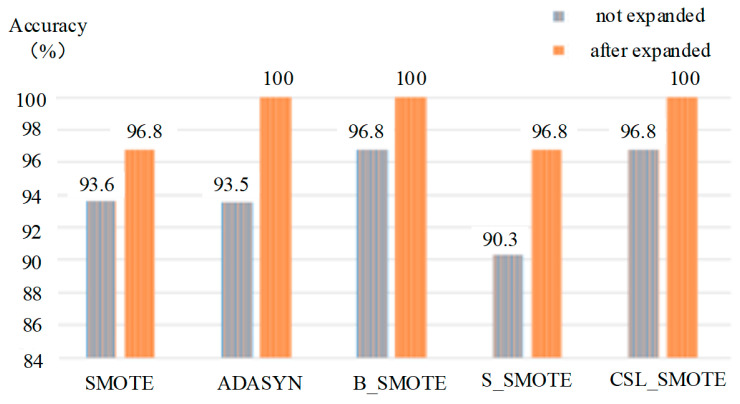
The classification of mixed gases based on SOFTMAX method.

#### 4.4.4. The Classification of Mixed Gases Based on MRVM Method

When the MRVM method is used to classify the mixed gas, the PSO and GA methods are used to optimize the parameters of the core parameter b in MRVM, and the kernel function is Gaussian, the optimization parameter is kernel parameter b. The results of the MRVM classification method as shown in Table 5 and Figure 9.

**Table 5 sensors-25-06254-t005:** The classification of mixed gases based on MRVM method.

MRVM	SMOTE	ADASYN	B_SMOTE	S_SMOTE	CSL_SMOTE
K	5	5	2	5	5
b (PSO)	59.7177	74.3593	93.5487	28.0625	4.7429
Unexpanded accuracy (%)	93.6	96.8	96.8	90.3	93.6
Expanded accuracy (%)	96.8	96.8	100	93.6	100
b (GA)	71.6119	27.9014	74.6933	35.3773	89.9384
Unexpanded accuracy (%)	96.8	93.6	96.8	90.32	96.8
Expanded accuracy (%)	100	100	96.8	100	97.8

**Figure 9 sensors-25-06254-f009:**
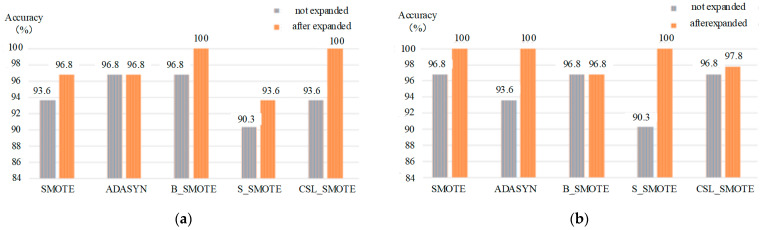
The classification of different methods: (**a**) The classification of gas mixtures based on the PSO-MRVM method; (**b**) the classification of mixed gases based on the GA-MRVM method.

#### 4.4.5. The Classification of Mixed Gases Based on SVM Method

When the SVM method was used to classify the gas mixture, the PSO and GA methods were used to optimize the parameters of kernel parameter g and penalty parameter C in SVM, and the kernel function is Gaussian. The results of the SVM classification method as shown in Table 6 and Figure 10.

**Table 6 sensors-25-06254-t006:** The classification of mixed gases based on SVM method.

SVM	SMOTE	ADASYN	B_SMOTE	S_SMOTE	CSL_SMOTE
K	5	5	2	5	5
C (PSO)	6.7118	14.0453	5.2927	9.8601	6.9446
g (PSO)	3.9934	1.7689	5.5391	4.8145	4.8246
Unexpanded accuracy (%)	93.5	93.5	93.5	96.8	93.5
Expanded accuracy (%)	100	96.8	96.8	96.8	96.8
C (GA)	24.0598	19.0548	3.7400	9.8601	99.5516
g (GA)	3.0456	4.3361	19.8891	0.8145	13.2976
Unexpanded accuracy (%)	96.88	93.5	93.75	96.88	96.8
Expanded accuracy (%)	96.88	100	100	96.88	96.8

**Figure 10 sensors-25-06254-f010:**
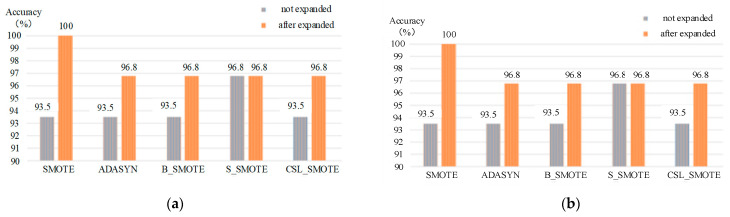
The classification of different methods: (**a**) The classification of mixed gases based on the PSO-SVM method; (**b**) the classification of mixed gases based on GA-SVM method.

#### 4.4.6. The Classification of Mixed Gases Based on ELM Method

When the ELM method is used to classify mixed gas, the PSO and GA methods are used to optimize parameters of weight w (8 × 16) and bias b (8 × 1) in ELM method. The results of ELM method are shown in Table 7 and Figure 11.

**Table 7 sensors-25-06254-t007:** The classification of mixed gases based on ELM method.

Methods	SMOTE	ADASYN	B_SMOTE	S_SMOTE	CSL_SMOTE
K	5	5	2	5	5
Unexpanded accuracy (%)	90.3	93.6	90.3	93.55	96.8
Expanded accuracy (%)	96.8	100	100	96.77	100
Unexpanded accuracy (%)	93.6	90.3	90.3	93.6	93.6
Expanded accuracy (%)	96.8	96.7	100	96.77	96.8

**Figure 11 sensors-25-06254-f011:**
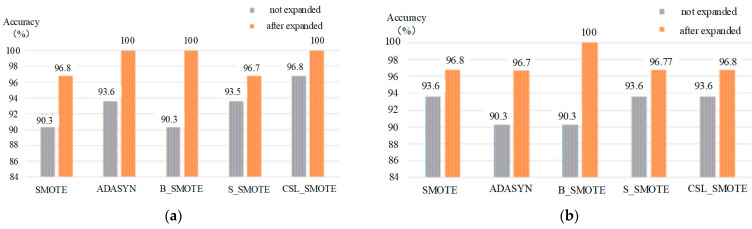
The classification of different methods: (**a**) The classification of mixed gases based on the PSO-ELM method; (**b**) the classification of mixed gases based on GA-ELM method.

#### 4.4.7. The Comparison of Performance Based on Different Method Classifications

When classifying gas mixture components, the PSO and GA methods are used to optimize the parameters of ELM, SVM, MRVM and SOFTMAX classification methods. In order to evaluate the difference in overall performance of the classification methods over the sample expansion method, the statistical average of the expansion difference under two optimization algorithms under the same sample expansion method is calculated (for example, SMOTE is the statistical average of PSO-ELM and GA-ELM under the same classification method). The extended mean difference in Figure 12 is the statistical average of the extended mean difference (SMOTE, ADASYN, B-SMOTE, S_SMOTE and CSL_SMOTE). The extended mean difference in ELM is 2.6% larger than that in SVM, 2.1% larger than that in MRVM, and 1% larger than that in SOFTMAX. Therefore, the extended mean difference in ELM is the largest, indicating that the classification method has the best overall performance in the proposed sample expansion method.(28)$$GA-SMOTE-{ELM}_{Extended\;difference}=GA\_SMOTE\_{ELM}_{Accuracy\;rate}-MAPE-GA\;\_ELM_{Accuracy\;rate}$$(29)$$SMOTE_{Extended\;difference}=(GA-SMOTE-ELM_{Extended\;difference}+PSO-SMOTE-ELM_{Extended\;difference})$$(30)$$\begin{array}{l} ELM_{Extended\;difference}=(SMOTE_{Extended\;difference}+ADASYN_{Extended\;difference}+B\_SMOTE_{Extended\;difference}+\\ S\_SMOTE_{Extended\;difference}+CSL\_SMOTE_{Extended\;difference}) \end{array}$$

It can be seen from Figure 8, Figure 9, Figure 10 and Figure 11 that different sample expansion methods have different performance in classification methods. In order to evaluate the overall difference in the performance of sample expansion methods in classification methods, the extended difference in two optimization algorithms under the same classification method is statistically averaged (for example, the extended difference (ELM) is the statistical average of PSO-ELM and GA-ELM under the same sample extension), and the extended mean difference in Figure 13 is the statistical average of the extended difference (ELM, SVM, MRVM and SOFTMAX). It can be seen from the figure that the expansion mean difference is greater than 0, and ADASYN’s expansion mean difference is greater than 0.1% of B_SMOTE, 1.4% of S_SMOTE, 1.7% of SMOTE, and 2.5% of CSL_SMOTE, so the ADASYN method has the largest expansion mean difference. It shows that the sample expansion method has the best overall performance among the proposed classification methods.(31)$$GA-SMOTE-SVR_{Extended\;difference}=GA\_SMOTE\_SVR_{Accuracy\;rate}-MAPE-GA\;\_SVR_{Accuracy\;rate}$$(32)$$SMOTESVR_{Extended\;difference}=\left(GA-SMOTE-SVR_{Extended\;difference}+PSO-SMOTE-SVR_{Extended\;difference}\right)/2$$(33)$$SMOTE_{Extended\;difference}=\left(ELM_{Extended\;difference}+SVM_{Extended\;difference}+MRVM_{Extended\;difference}+SOFTMAX_{Extended\;difference}\right)/4$$

### 4.5. The Quantitative Analysis

In the quantitative analysis of the mixed gas, PSO and GA are used to optimize the corresponding parameters in the classification model, and the optimal parameter classification model is obtained to estimate the concentration of the mixed gas. In order to evaluate the effectiveness of this method, appropriate evaluation indexes are needed. In this paper, mean absolute percentage error (MAPE) and root mean square error (RMSE) are selected as evaluation indexes of the prediction model, as shown in Equations (34) and (35). In Table 8, Table 9, Table 10, Table 11, Table 12, Table 13, Table 14, Table 15 and Table 16, the MAPE and RMSE of the sample of the gas mixing part increased by sample expansion were both lower than those before sample expansion, indicating that the sample expansion method is beneficial to concentration estimation of the class imbalance data set.(34)$$MAPE=\frac{1}{n}\sum_{i=1}^{n}\left|\frac{{\hat{y}}_{i}-y_{i}}{y_{i}}\right|\times 100\%$$(35)$$RMSE=\sqrt{\frac{\sum_{i=1}^{n}{\left({\hat{y}}_{i}-y_{i}\right)}^{2}}{N}}$$
where $y_{i}$ is the true concentration value, ${\hat{y}}_{i}$ is the predicted concentration value, and n is the number of samples to be measured.

#### 4.5.1. The Quantitative Analysis of Mixed Gases Based on SVR Method

When the SVR method is used to the quantitative analysis of mixed gas, the PSO and GA methods are used to optimize the parameters of penalty coefficient C, nuclear parameter g in SVR and the kernel function is Gaussian. The results of the quantitative analysis of mixed gases based on SVR method as shown in Table 8, Table 9 and Table 10.

**Table 8 sensors-25-06254-t008:** The estimation of mixed gas concentration based on the parameter-optimal SMOTE-SVR model.

	Single Gas			Mixed Gas			
Number of tests	15	34	18	8		5	
Mixed sample size	0	0	0	25		29	
Gas composition	CO	Eth.	Met.	Eth.	CO	Eth.	Met.
C (PSO)	33.2384	100	17.4843	50.5712	81.6711	5.0525	15.0712
g (PSO)	0.01	0.0817	0.01	1.0522	1.0678	0.1	7.5242
RMSE (unexpanded)	0.3336	11.8565	0.382	59.0951	1.2374	7.7482	1.3761
RMSE (expanded)	\	\	\	16.1838	0.3614	6.1206	0.6009
MAPE (unexpanded)	0.0267	0.0342	0.0269	0.175	0.1088	0.0653	0.1585
MAPE (expanded)	\	\	\	0.0464	0.0269	0.0424	0.0458
C (GA)	27.1136	99.5452	46.3705	8.6435	99.1343	87.3844	17.1865
g (GA)	0.0422	0.2859	0.037	0.9762	0.3067	0.0809	18.3491
RMSE (unexpanded)	0.2906	12.1342	0.2717	60.1283	0.9031	7.7985	1.4123
RMSE (expanded)	\	\	\	16.6925	0.1512	10.0719	0.904
MAPE (unexpanded)	0.0202	0.0272	0.0114	0.1751	0.0932	0.0736	0.1626
MAPE (expanded)	\	\	\	0.0496	0.0126	0.0677	0.0942

**Table 9 sensors-25-06254-t009:** The estimation of mixed gas concentration based on the parameter-optimal ADASYN-SVR model.

	Single Gas			Mixed Gas			
Number of tests	15	34	18	8		5	
Mixed sample size	0	0	0	25		29	
Gas composition	CO	Eth.	Met.	Eth.	CO	Eth.	Met.
C (PSO)	33.2384	100	17.4843	78	60.3982	77.906	12.4882
g (PSO)	0.01	0.0817	0.01	0.1	0.1	1.3654	0.2321
RMSE (unexpanded)	0.3336	11.8565	0.382	70.2749	0.9247	7.7482	1.3761
RMSE (expanded)	\	\	\	33.6168	0.3545	6.1206	0.6009
MAPE (unexpanded)	0.0267	0.0342	0.0269	0.1721	0.0937	0.2867	0.0501
MAPE (expanded)	\	\	\	0.0686	0.0167	0.0558	0.0082
C (GA)	27.1136	99.5452	46.3705	31.5813	43.7329	90.7568	46.2463
g (GA)	0.0422	0.2859	0.037	0.9524	0.3248	0.3658	0.1222
RMSE (unexpanded)	0.2906	12.1342	0.2717	60.4922	0.906	7.7482	1.3761
RMSE (expanded)	\	\	\	43.391	0.1591	6.1206	0.6009
MAPE (unexpanded)	0.0202	0.0272	0.0114	0.1752	0.0932	0.1127	0.0841
MAPE (expanded)	\	\	\	0.0971	0.0101	0.0514	0.0122

**Table 10 sensors-25-06254-t010:** The estimation of mixed gas concentration based on the parameter-optimal S_SMOTE-SVR model.

	Single Gas			Mixed Gas			
Number of tests	15	34	18	8		5	
Mixed sample size	0	0	0	25		29	
Gas composition	CO	Eth.	Met.	Eth.	CO	Eth.	Met.
C (PSO)	33.2384	52.8475	17.4843	100	42.4828	100	22.3426
g (PSO)	0.01	1.1183	0.01	0.1586	0.0694	0.01	0.2
RMSE (unexpanded)	0.3336	11.8565	0.382	70.43	0.9518	7.0667	0.5797
RMSE (expanded)	\	\	\	4.2233	0.5892	6.5782	0.5106
MAPE (unexpanded)	0.0267	0.0342	0.0269	0.1757	0.0936	0.0677	0.0662
MAPE (expanded)	\	\	\	0.0101	0.0369	0.0534	0.0596
C (GA)	27.1136	99.5452	46.3705	38.6996	53.4205	100	53.0962
g (GA)	0.0422	0.2859	0.037	0.5453	1.1621	0.3564	2.9062
RMSE (unexpanded)	0.2906	12.1342	0.2717	68.2089	1.2667	15.8155	1.1962
RMSE (expanded)	\	\	\	0.0135	0.9363	5.2303	0.6657
MAPE (unexpanded)	0.0202	0.0272	0.0114	0.1724	0.1119	0.1097	0.1347
MAPE (expanded)	\	\	\	0.0135	0.0544	0.049	0.0472

#### 4.5.2. The Quantitative Analysis of Mixed Gases Based on ELM Method

When the ELM method is used to the quantitative analysis of mixed gas, the PSO and GA methods are used to optimize the parameters of weight W (8 × 16) and bias b (8 × 1) in ELM. The results of the quantitative analysis of mixed gases based on ELM method are shown in Table 11, Table 12 and Table 13.

**Table 11 sensors-25-06254-t011:** The estimation of mixed gas concentration based on the parameter-optimal SMOTE-ELM model.

	Single Gas			Mixed Gas			
Number of tests	15	34	18	8		5	
Mixed sample size	0	0	0	25		29	
Gas composition	CO	Eth.	Met.	Eth.	CO	Eth.	Met.
unexpanded RMSE (GA)	0.0779	7.5928	0.0834	4.2883	0.7327	1.5992	1.1006
expanded RMSE (GA)	\	\	\	0.5538	0.7063	5.4086	0.6163
unexpanded MAPE (GA)	0.0072	0.0242	0.0068	0.0135	0.065	0.0125	0.1345
expanded MAPE (GA)	\	\	\	0.0012	0.0275	0.0304	0.066
unexpanded RMSE (PSO)	0.1059	8.0137	0.0699	2.5519	1.2904	2.215	1.0358
expanded RMSE (PSO)	\	\	\	1.4328	0.3904	1.6374	0.7018
unexpanded MAPE (PSO)	0.0056	0.0265	0.0036	0.0063	0.112	0.013	0.1288
expanded MAPE (PSO)	\	\	\	0.0041	0.0246	0.0126	0.0694

**Table 12 sensors-25-06254-t012:** The estimation of mixed gas concentration based on the parameter-optimal ADASYN-ELM model.

	Single Gas			Mixed Gas			
Number of tests	15	34	18	8		5	
Mixed sample size	0	0	0	25		29	
Gas composition	CO	Eth.	Met.	Eth.	CO	Eth.	Met.
unexpanded RMSE (GA)	0.0889	7.2484	0.1110	5.748	0.8527	2.4868	1.0435
expanded RMSE (GA)	\	\	\	2.3989	0.8407	2.5609	0.9546
unexpanded MAPE (GA)	0.0063	0.0212	0.0078	0.0189	0.0826	0.0151	0.1264
expanded MAPE (GA)	\	\	\	0.0149	0.0614	0.021	0.0846
unexpanded RMSE (PSO)	0.0937	6.6633	0.0505	3.0731	1.4314	1.4958	0.7338
expanded RMSE (PSO)	\	\	\	3.4044	1.1403	1.6688	0.5138
unexpanded MAPE (PSO)	0.0067	0.0217	0.0029	0.0079	0.1183	0.0135	0.0916
expanded MAPE (PSO)	\	\	\	0.0078	0.0631	0.0118	0.0448

**Table 13 sensors-25-06254-t013:** The estimation of mixed gas concentration based on the parameter-optimal S_SMOTE-ELM model.

	Single Gas			Mixed Gas			
Number of tests	15	34	18	8		5	
Mixed sample size	0	0	0	25		29	
Gas composition	CO	Eth.	Met.	Eth.	CO	Eth.	Met.
unexpanded RMSE (GA)	0.0774	8.2827	0.1082	2.9036	1.0441	2.8884	1.0022
expanded RMSE (GA)	\	\	\	0.0785	0.0176	2.5923	0.4906
unexpanded MAPE (GA)	0.005	0.0254	0.0085	0.0083	0.0926	0.0208	0.1202
expanded MAPE (GA)	\	\	\	0.0001	0.0008	0.0188	0.0465
unexpanded RMSE (PSO)	0.0902	6.325	0.0059	1.6848	1.0424	2.2192	0.7259
expanded RMSE (PSO)	\	\	\	0.1514	0.18	2.1365	0.3481
unexpanded MAPE (PSO)	0.0074	0.0189	0.0852	0.0051	0.1038	0.0118	0.087
expanded MAPE (PSO)	\	\	\	0.0003	0.0112	0.0128	0.0329

#### 4.5.3. The Quantitative Analysis of Mixed Gases Based on MLSSVR SVR Method

When the MLSSVR method is used to the quantitative analysis of mixed gas, the PSO and GA methods are used to optimize the parameters of gamma, lambda and p in MLSSVR, and the kernel function is Gaussian. The results of the quantitative analysis of mixed gases based on MLSSVR method are shown in Table 14, Table 15 and Table 16.

**Table 14 sensors-25-06254-t014:** The estimation of mixed gas concentration based on the parameter-optimal SMOTE-MLSSVR model.

	Single Gas			Mixed Gas			
Number of tests	15	34	18	8		5	
Mixed sample size	0	0	0	25		29	
Gas composition	CO	Eth.	Met.	Eth.	CO	Eth.	Met.
Gamma (PSO)	20.3199	132.6191	7.8837	15.3982		71.6237	
Lambda (PSO)	0.01	0.01	0.01	0.1		0.1	
P (PSO)	0.051	0.2146	0.0399	0.7856		0.0364	
unexpanded RMSE	0.2796	11.1042	0.2467	33.9165	1.2903	37.5702	4.4949
expanded RMSE	\	\	\	14.2842	0.6261	10.1838	0.6122
unexpanded MAPE	0.019	0.0268	0.0146	0.0941	0.1074	0.2846	0.5559
expanded MAPE	\	\	\	0.036	0.0359	0.054	0.0681
Gamma (GA)	99.9857	45.3705	99.4991	15.3982		97.2574	
Lambda (GA)	0.0731	69.4503	0.1449	0.1		0.0107	
P (GA)	0.0547	0.1193	0.0395	0.7856		0.0194	
unexpanded RMSE	0.2784	16.5005	0.2468	33.9363	1.2845	17.9652	0.9144
expanded RMSE	\	\	\	13.6607	1.3361	6.6101	0.3365
unexpanded MAPE	0.0189	0.0559	0.0145	0.0943	0.1066	0.1362	0.1031
expanded MAPE	\	\	\	0.0344	0.0761	0.0385	0.0369

**Table 15 sensors-25-06254-t015:** The estimation of mixed gas concentration based on the parameter-optimal ADASYN-MLSSVR model.

	Single Gas			Mixed Gas			
Number of tests	15	34	18	8		5	
Mixed sample size	0	0	0	25		29	
Gas composition	CO	Eth.	Met.	Eth.	CO	Eth.	Met.
Gamma (PSO)	20.3199	132.6191	7.8837	1.4685		100	
Lambda (PSO)	0.01	0.01	0.01	0.1		0.1	
P (PSO)	0.051	0.2146	0.0399	12.2774		0.1203	
unexpanded MAPE	0.2796	11.1042	0.2467	0.0941	0.1074	0.1898	0.4132
expanded MAPE	\	\	\	0.1181	0.0785	0.0781	0.0574
unexpanded RMSE	0.019	0.0268	0.0146	33.9165	1.2903	27.1887	3.3801
expanded RMSE	\	\	\	42.4597	1.2937	11.8	0.4922
Gamma (GA)	99.9857	45.3705	99.4991	15.3982		96.3928	
Lambda (GA)	0.0731	69.4503	0.1449	0.1		9.5367	
P (GA)	0.0547	0.1193	0.0395	0.7856		0.0188	
unexpanded MAPE	0.2784	16.5005	0.2468	0.0943	0.1065	0.1363	0.0369
expanded MAPE	\	\	\	0.1188	0.1814	0.0373	0.033
unexpanded RMSE	0.0189	0.0559	0.0145	33.9367	1.2843	17.9737	0.3623
expanded RMSE	\	\	\	42.7694	1.8754	4.8422	0.3491

**Table 16 sensors-25-06254-t016:** The estimation of mixed gas concentration based on the parameter-optimal S_SMOTE-MLSSVR model.

	Single Gas			Mixed Gas			
Number of tests	15	34	18	8		5	
Mixed sample size	0	0	0	25		29	
Gas composition	CO	Eth.	Met.	Eth.	CO	Eth.	Met.
Gamma (PSO)	20.3199	132.6191	7.8837	100		100	
Lambda (PSO)	0.01	0.01	0.01	0.1		0.1	
P (PSO)	0.051	0.2146	0.0399	3.06292		0.1778	
unexpanded RMSE	0.2796	11.1042	0.2467	33.9141	1.2739	27.1887	3.3801
expanded RMSE	\	\	\	3.06292	0.0575	8.9217	0.7676
unexpanded MAPE	0.019	0.0268	0.0146	0.0947	0.1063	0.1898	0.4132
expanded MAPE	\	\	\	0.0076	0.0032	0.0789	0.0949
Gamma (GA)	99.9857	45.3705	99.4991	99.8776		95.6688	
Lambda (GA)	0.0731	69.4503	0.1449	0.0011		0.0203	
P (GA)	0.0547	0.1193	0.0395	0.5661		0.0869	
unexpanded MAPE	0.2784	16.5005	0.2468	0.0943	0.1067	0.1362	0.0671
expanded MAPE	\	\	\	0.0044	0.0009	0.08	0.0369
unexpanded RMSE	0.0189	0.0559	0.0145	33.9331	1.2847	17.9612	0.6
expanded RMSE	\	\	\	2.2973	0.0238	9.8236	0.3366

#### 4.5.4. The Comparison of Performance Based on Different Quantitative Analysis Methods

It can be seen from the above table that MAPE and RMSE after sample expansion are generally lower than those before sample expansion, indicating that the sample expansion method is effective in estimating the concentration of mixed gas. In order to highlight the positive effect of the sample expansion method on various classification methods, the difference between MAPE and RMSE after sample expansion and before sample expansion is used. And the larger the difference before and after expansion, the better the effect of the expanded sample method, as shown in Figure 14 and Figure 15. In Figure 14, under the MAPE evaluation standard, SMOTE sample extension method has the best performance on PSO-SVR, and S-SMOTE sample extension method has the best performance on PSO-ELM and GA-MLSSVR. In Figure 15, under the RMSE evaluation standard, SMOTE sample extension method has the best performance on PSO-SVR, and S_SMOTE sample extension method has the best performance on PSO-ELM and PSO-MLSSVR.

As shown in Figure 14 and Figure 15 above, the same classification method has different performance for different sample expansion methods. In order to find the best sample expansion method conducive to classification, the expansion difference in two optimization algorithms under the same classification method is statistically averaged, as shown in Table 17 and Table 18. For example, the extended difference (SVR) in Table 11 is the statistical average of PSO-SVR and GA-SVR under the same sample expansion, and the extended mean difference in the table is the statistical average of the extended difference (SVR, ELM and MLSSVR). Table 18 is similar. As can be seen from Figure 16a, under the MAPE evaluation standard, S_SMOTE SMOTE average difference in expansion is greater than that of ADASYN by 5.85% and SMOTE by 9.43%. The expansion mean difference in ADASYN is 3.58% larger than that of SMOTE, so S_SMOTE has the best overall performance. As can be seen from Figure 16b, under the RMSE evaluation standard, S_SMOTE average difference in expansion is 2.6% larger than ADASYN average difference in expansion, and 8.67% larger than SMOTE average difference in expansion. Compared with ADASYN, SMOTE expansion mean difference is 6.07% larger, so S-SMOTE overall performance is the best. Finally, under the two evaluation criteria, S-SMOTE sample expansion method is the best classification method.(36)$$GA-SMOTE-SVR_{expansion\;difference}=GA-SVR\_MAPE-GA-SMOTE-SVR\_MAPE$$(37)$$SVR_{expansion\;difference}=\left(GA-SMOTE-SVR_{expansion\;difference}+PSO-SMOTE-SVR_{expansion\;difference}\right)/2$$(38)$$SMOTE_{expansion\;difference}=\left(SVR_{expansion\;difference}+ELM_{expansion\;difference}+MLSSVR_{expansion\;difference}\right)/3$$

In order to more clearly highlight the positive effect of sample expansion methods on various classification methods, the difference between MAPE and RMSE after sample expansion and before sample expansion is used. The larger the difference before and after sample expansion, the better the performance of regression method under the same sample expansion, as shown in Figure 17 and Figure 18. In Figure 17, under the MAPE evaluation criteria, PSO-MLSSVR regression method has the best performance on SMOTE and ADASYN sample expansion methods, and GA-MLSSVR has the best performance on S_SMOTE method. In Figure 18, under the RMSE evaluation criteria, the PSO-MLSSVR classification method has the best performance on SMOTE and S_SMOTE methods, and the PSO-SVR method has the best performance on ADASYN method.

As shown in Figure 17 and Figure 18 above, different classification methods differ in the performance of the same sample expansion method. In order to find the best classification method based on sample expansion, the statistical average of the expansion difference in the two optimization algorithms under the same sample expansion method is conducted, as shown in Table 19 and Table 20. For example, the extended difference in Table 18 is the statistical average of PSO-SVR and GA-SVR under the same regression method, and the extended mean difference in Table 18 is the statistical average of the extended difference (SMOTE, ADASYN and S_SMOTE). Table 20 is similar. It can be seen from Figure 19a that under MAPE evaluation criteria, MLSSVR’s average difference in expansion is 26.04% larger than SVR’s average difference in expansion, and 33.09% larger than ELM’s average difference in expansion. The extended mean difference in SVR is 7.05% larger than that of ELM, so MLSSVR has the best overall performance. It can be seen from Figure 19b that under RMSE evaluation criteria, SVR’s average difference in expansion is 12.53% larger than MLSSVR’s average difference in expansion and 12.53% larger than ELM’s average difference in expansion. MLSSVR’s extended mean difference is 11.84% larger than ELM’s extended mean difference, so SVR has the best overall performance. However, MLSSVR classification method is slightly better than SVR classification method in sample expansion under the two evaluation criteria.(39)$$GA-SMOTE-SVR_{expansion\;difference}=GA-SVR\_MAPE-GA-SMOTE-SVR\_MAPE$$(40)$$SVR_{expansion\;difference}=\left(GA-SMOTE-SVR_{expansion\;difference}+PSO-SMOTE-SVR_{expansion\;difference}\right)/2$$(41)$$SVR_{expansion\;difference}=\left(SMOTE_{expansion\;difference}+ADASYN_{expansion\;difference}+S\_SMOTE_{expansion\;difference}\right)/3$$

From the above results of concentration estimation, it can be seen that the sample expansion method and classification method proposed in this paper are effective in estimating the concentration of mixed gas. Figure 20 shows the estimation of mixed gas concentration based on the PSO-S_SMOTE_MLSSVR method. It can be seen that S_SMOTE method was used to combine artificial samples. Therefore, the concentration estimation method based on sample expansion is better.

## 5. Conclusions

The present work has been an attempt to propose a method of composition identification and concentration estimation of mixed gases based on sample expansion to alleviate the problem of extremely unbalanced gas sample number and too few samples. The ADASYN-ELM gas identification method is proposed for the qualitative identification of mixed gas components. In this method, KPCA is used to extract the characteristic value of the sensor array signal and obtain the characteristic value of the gas mixture component, which solves the problem that the response of the MOS gas sensor to the gas is nonlinear. Then, the ADASYN method is used to expand the sample to solve the problem of too few samples. The PSO and GA optimization algorithms were used to optimize the parameters of ELM classification method, solve the problem of multi-parameter and difficult parameter determination in the model, then obtain the optimal parameter classification model for qualitative analysis of mixed gas. For quantitative estimation of mixed gases concentration, S-SMOTE-MLSSVR mixed gases concentration estimation method was put forward. First, the sample expansion method S-SMOTE was used to expand the sample, and the parameters of the regression method MLSSVR were optimized by PSO and GA optimization algorithms, and the regression model with the best parameters were obtained. In order to verify the effectiveness of the proposed method, a public data set is used for experimental verification. The results show that for the classification part, the accuracy rate after sample expansion is generally higher than before sample expansion. For the quantitative estimation part, the MAPE and RMSE after sample expansion are generally lower than before sample expansion, indicating that the sample expansion method has a positive effect on the classification and concentration estimation of mixed gases with extremely unbalanced and too few samples.

## Figures and Tables

**Figure 1 sensors-25-06254-f001:**
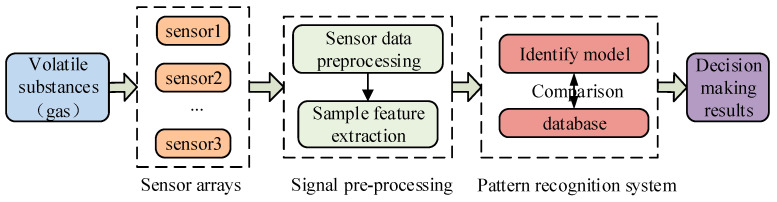
The workflow of electronic nose system.

**Figure 2 sensors-25-06254-f002:**
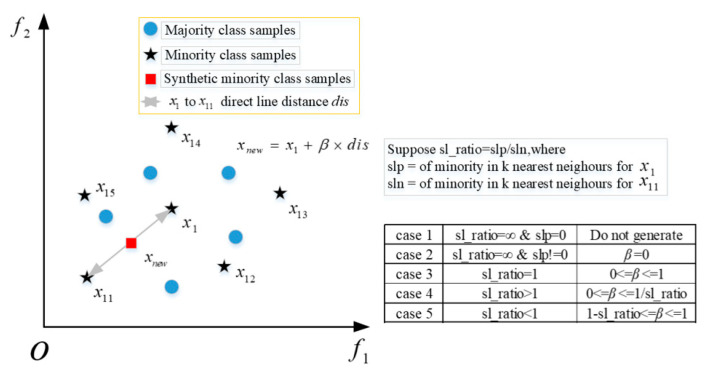
The diagram of Safe-Level-SMOTE synthesis of minority class samples.

**Figure 3 sensors-25-06254-f003:**
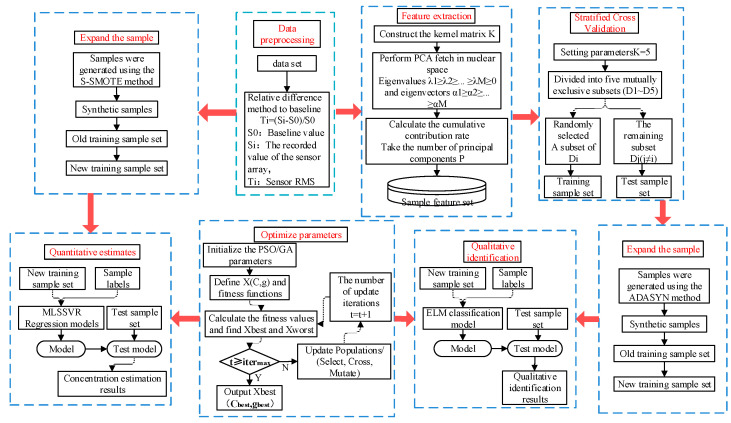
The mixture gas identification and concentration detection method based on sample expansion.

**Figure 4 sensors-25-06254-f004:**
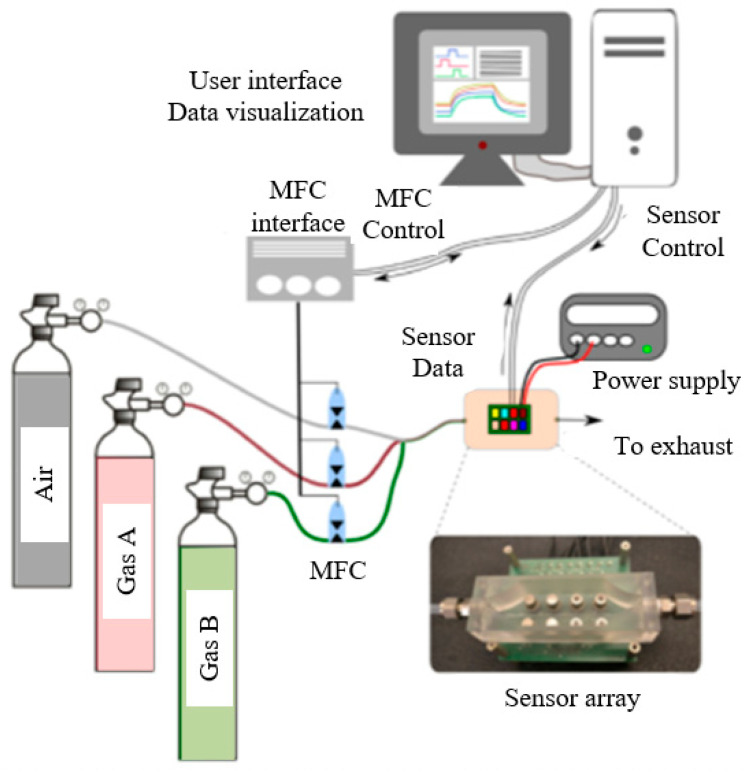
Experimental system diagram.

**Figure 5 sensors-25-06254-f005:**
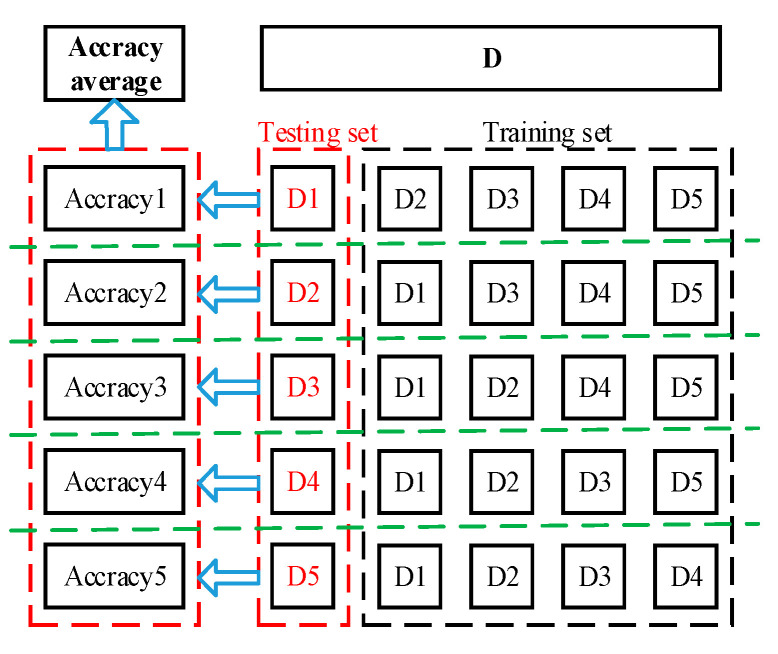
The five-fold hierarchical cross-validation data allocation.

**Figure 6 sensors-25-06254-f006:**
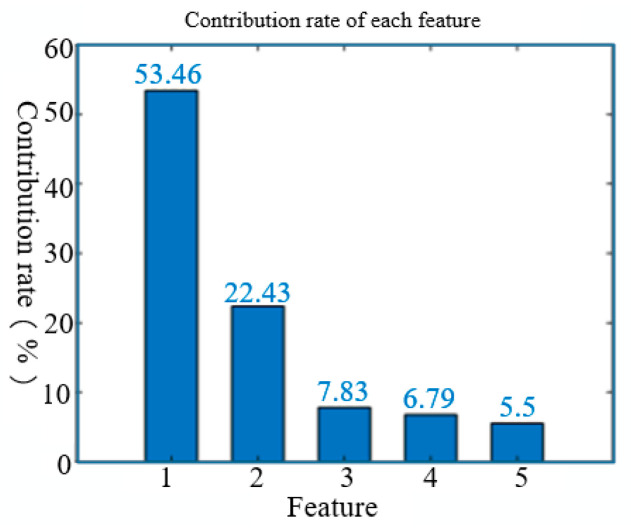
The contribution rate arranged in the histogram.

**Figure 7 sensors-25-06254-f007:**
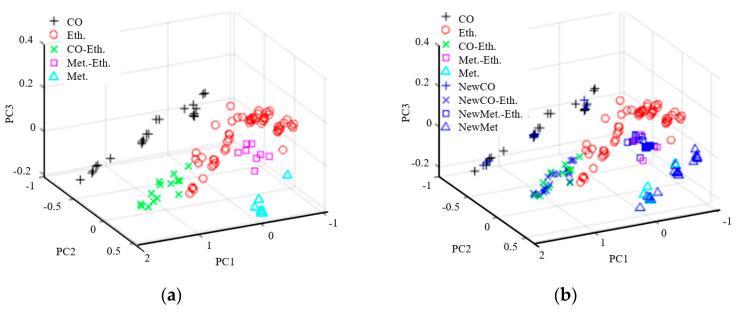
The sample distribution maps of different methods: (**a**) The distribution of various data sets with the first three principal components; (**b**) the synthetic sample with S-SMOTE method.

**Figure 12 sensors-25-06254-f012:**
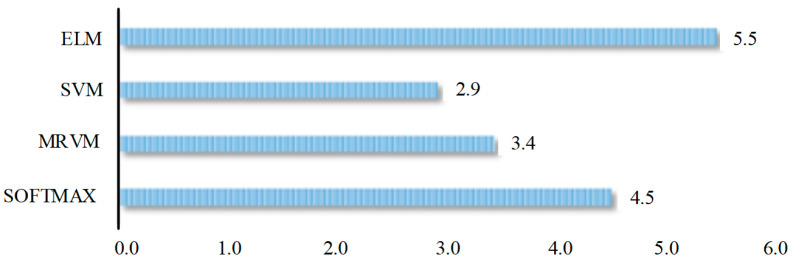
The comparison of performance based on different method classifications.

**Figure 13 sensors-25-06254-f013:**
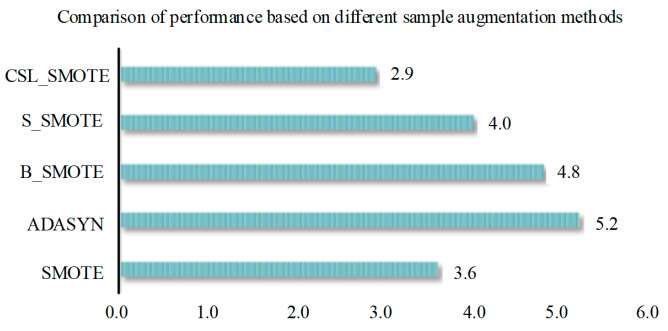
The comparison of performance based on different sample augmentation methods.

**Figure 14 sensors-25-06254-f014:**
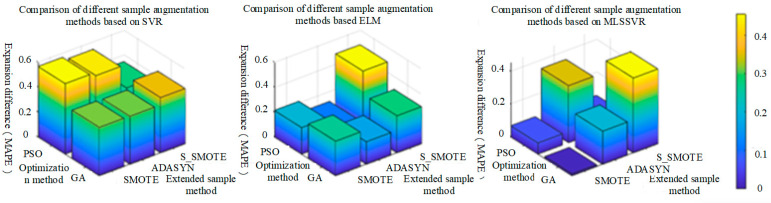
The performance of different sample expansion methods based on MAPE evaluation criteria.

**Figure 15 sensors-25-06254-f015:**
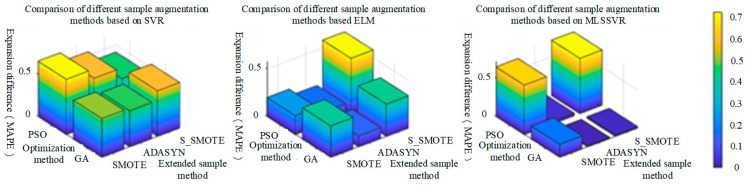
The performance of different sample expansion methods based on RMSE evaluation criteria.

**Figure 16 sensors-25-06254-f016:**
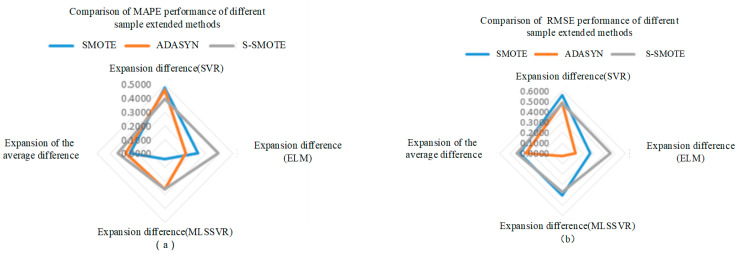
The comparison of MAPE and RMSE performance of different sample expansion methods.

**Figure 17 sensors-25-06254-f017:**
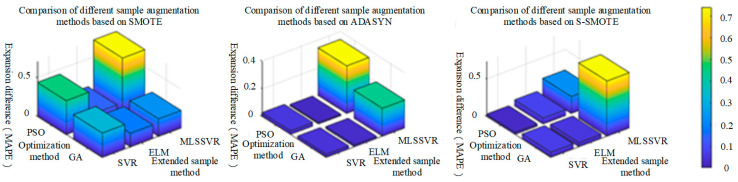
The performance of different classification methods based on MAPE evaluation criteria.

**Figure 18 sensors-25-06254-f018:**
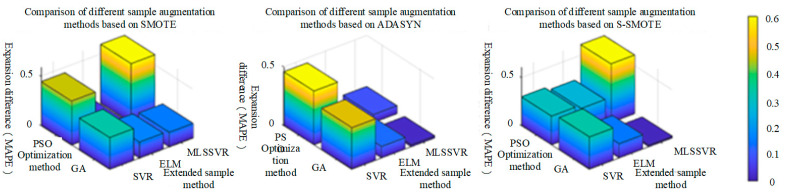
The performance of different classification methods based on RMSE evaluation criteria.

**Figure 19 sensors-25-06254-f019:**
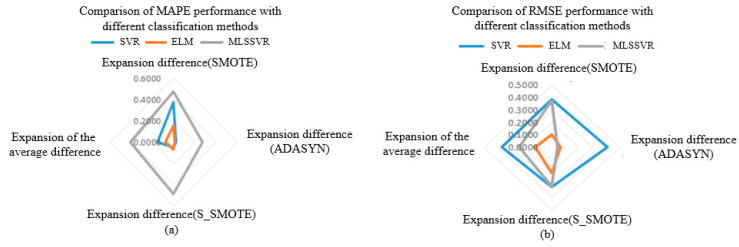
The comparison of MAPE and RMSE performance of different classification methods.

**Figure 20 sensors-25-06254-f020:**
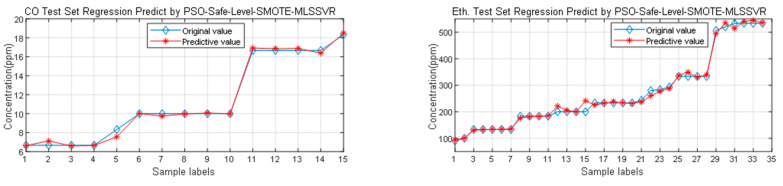
The sample distribution maps of different methods.

**Table 1 sensors-25-06254-t001:** The data set introduction.

Label	Gas	Sample Size
1	CO	30
2	Ethylene	68
3	CO–Ethylene	17
4	Methane–Ethylene	10
5	Methane	36

**Table 2 sensors-25-06254-t002:** The expansion amounts of each gas based on different sample expansion methods for qualitative analysis.

Label	Gas	SMOTE	ADASYN	B_SMOTE	S_SMOTE	CSL_SMOTE
1	CO	0	0	0	0	13
2	Eth.	0	0	0	0	0
3	Met.	0	0	0	0	23
4	CO and Eth.	25	23	22	25	25
5	Met. and Eth.	29	28	28	29	29

**Table 3 sensors-25-06254-t003:** The expansion amounts of each gas based on different sample expansion methods for concentration analysis.

Label	Gas	SMOTE	ADASYN	S_SMOTE
1	CO	0	0	0
2	Eth.	0	0	0
3	Met.	0	0	0
4	CO and Eth.	25	27	30
5	Met. and Eth.	29	34	35

**Table 17 sensors-25-06254-t017:** The comparison of MAPE performance of different sample expansion methods.

Methods	SMOTE	ADASYN	S_SMOTE
expansion difference (SVR)	0.4762	0.4614	0.3960
expansion difference (ELM)	0.2448	0.1508	0.3906
expansion difference (MLSSVR)	0.0403	0.2566	0.2578
expansion mean difference	0.2538	0.2896	**0.3481**

**Table 18 sensors-25-06254-t018:** The comparison of RMSE performance of different sample expansion methods.

Methods	SMOTE	ADASYN	S_SMOTE
expansion difference (SVR)	0.5573	0.4860	0.4776
expansion difference (ELM)	0.2635	0.1208	0.4586
expansion difference (MLSSVR)	0.4054	0.0236	0.3681
expansion mean difference	0.4088	0.3481	**0.4348**

**Table 19 sensors-25-06254-t019:** The comparison of MAPE performance of different classification methods.

Methods	SVR	ELM	MLSSVR
expansion difference (SMOTE)	0.3774	0.1580	0.4722
expansion difference (ADASYN)	0.0235	0.0116	0.2735
expansion difference (S_SMOTE)	0.0473	0.0672	0.4836
expansion mean difference	0.1494	0.0789	**0.4098**

**Table 20 sensors-25-06254-t020:** The comparison of RMSE performance of different classification methods.

Methods	SVR	ELM	MLSSVR
expansion difference (SMOTE)	0.3841	0.1055	0.3736
expansion difference (ADASYN)	0.4052	0.0638	0.0426
expansion difference (S_SMOTE)	0.3162	0.2051	0.3134
expansion mean difference	**0.3685**	0.1248	0.2432

## Data Availability

The original contributions presented in this study are included in the article. Further inquiries can be directed to the corresponding author.

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
