# Peer review of "A Mixed Gas Component Identification and Concentration Estimation Method for Unbalanced Gas Sensor Array Samples"

_sensors, 2025, doi:10.3390/s25196254_

Round 1

Reviewer 1 Report

Comments and Suggestions for Authors

The paper proposes a mixed gas component identification and concentration estimation method for unbalanced gas sensor array samples by combining ADASYN-ELM for qualitative classification and S-SMOTE-MLSSVR for quantitative regression, with KPCA for feature extraction and PSO/GA for parameter optimization. Experimental results on UCI gas datasets show that sample expansion significantly improves classification accuracy and reduces estimation errors compared to unexpanded datasets. Here are some comments according to this paper:

1. The paper addresses the important issue of unbalanced datasets in gas sensor array analysis. However, the novelty compared to prior works on SMOTE/ADASYN for sensor data needs clearer justification. 
2. The manuscript frequently compares expanded vs. unexpanded samples, but it would strengthen the novelty if the authors compared against more advanced recent deep learning methods (e.g., CNNs, RNNs) applied directly to unbalanced gas datasets.
3. KPCA is used for feature extraction, but the choice of kernel function and its parameters is not clearly discussed.
4. When you discuss the electronic nose with PCA, refer to relevant references such as DOI: 10.1016/j.snb.2022.131418.
5. The use of PSO and GA for parameter optimization is justified, but there is no discussion of computational cost. 
6. The reported classification accuracies (up to 100%) seem unusually high for such a small and noisy dataset. Could this be due to overfitting? A more cautious interpretation is recommended.
7. Figures 7–10 contain redundant information and could be consolidated for clarity. 
8. The study is limited to binary mixtures of gases (ethylene–methane, ethylene–CO). How would the method perform on more complex mixtures (3+ gases) or real-world multi-source environments?
9. The robustness of the method to sensor drift, noise, or cross-sensitivity is not evaluated. 

Reviewer 2 Report

Comments and Suggestions for Authors

This study is related to the implementation of the technology of an electronic nose combined with sample expansion algorithms like ADASYN and S-SMOTE, and machine learning protocols (MLSSVR) for component identification in a gas mixture. The applied method and general concept of this research look quite interesting and significant for further e-nose uses as a tool for chemical analysis and data interpretation. However, through the review of this manuscript, I would like to note some critical points whose elimination helps to improve the study’s quality.

Introduction section:

The introduction includes a huge number of references to studies about the uses of machine learning and pre-processing algorithms for a broad range of e-nose applications (lines 74 - 145). This expands this section a lot, and through all of these references, the ability of the reader to understand the essence becomes extremely difficult. Propose that the authors should summarize this section more compactly with a short review about the challenges and pros and cons of each of the studies, and make a logical path through all of the mentioned algorithms to the study’s hypothesis.

Moreover, the author underlined that there is a strong demand to improve the current machine learning algorithms for e-nose implementation. Although there are no notes about other problems and limitations related to the e-nose technology, such as the sensitivity of sensing materials to target gases and analytes, the reproducibility of sensor arrays, limitations in the quantity of discriminated analytes, etc. I suggest including the notes about these issues in the text of the manuscript, accompanied by suitable references.

Methods and Methodology section:

The authors include detailed descriptions and explanations about the pre-processing, data augmentation, and pattern recognition algorithms. However, it seems reasonable to add more information about the UCI publicly available data sets, i.e., what sensing materials were? How many sensors were used? What was the design of an e-nose? What about the signal generation and stability? Additional descriptions of these topics would demonstrate the full picture for readers.

The Experiment and Results:

Through analyzing the results of the study, I had some questions and concerns:

(1) The author did not demonstrate the difference between B_SMOTE, S_SMOTE, CSL_SMOTE, and SMOTE. Therefore, it seems complicated to analyze the data using such types of algorithms for the readers.

(2) What is the purpose of using the ADASYN algorithm for the qualitative analysis and the S-SMOTE sample expansion method for the quantitative one? Why these methods, and what are the advantages of their use for the sample expansion compared to the alternative approaches?

(3) In Figure 6, there is no explained variance of each principal component.

(4) Why are PSO and GA needed for the data optimization? What is the main function of these algorithms?

(5) According to Table 1, the number of samples is quite small, and data expansion algorithms do not look sufficient to increase them greatly. How is the accuracy and efficiency of the proposed approaches for large data samples (1000 data points and more)? This is extremely essential for the applicability of the approaches for further integration to real uses of e-noses.

(6) The comparison of the proposed methods with alternative ones help to better estimate the contribution of this study for analyzing the data sets collected by multisensor arrays.

Reviewer 3 Report

Comments and Suggestions for Authors

Line 11: Please define KPCA the first time you use the abbreviation.

Line 27:  Replace colorimetry with spectroscopy

Line 31:  The sentence is difficult to understand. Please rewrite.

Line 45 "array is a single sensor": what do you mean?

Line 147: "when the number of samples is extremely unbalanced"  What do you mean by an unbalanced gas mixture?

Comments on the Quality of English Language

Line 26: the human body, not human body

Line 36: remove the word speed

Round 2

Reviewer 2 Report

Comments and Suggestions for Authors

I appreciate the authors for their improvements and modifications in the final version of the manuscript. However, I suggest taking into account several minor points that must be included:

Introduction section:

(1) Please double check the numeration of the references in the introduction of the manuscript, e.g., lines 32-36; references 60-62 are going in the beginning of the manuscript.
(2) Additionally, the authors discussed the problems and issues related to the implementation of single gas sensors and referred to the papers [60-62]. These issues seem quite obvious to the readers. I suggest adding some details about the current problems with e-nose use, which I have mentioned in the previous review report. There are several studies related to these issues made by Dr. Fedor S. Fedorov, Prof. Victor Sysoev, Prof. Alexander Sinitskii, and others. Please review their studies and research, and add some references.

Results and Discussion section:

(1) Currently, the description of the results and discussion looks better. However, I suggest adding a comparison overview of the proposed approach with other AI protocols related to e-nose data analysis and dataset sample expansion according to the literature sources and references.

(2) Please change the term 'a mixed gas' to 'a gas mixture' or 'a gas mixture component' (lines 8, 11, 24, 83, 102, 286, etc). "A mixed gas" does not seem correct and well understood.
